# Quantifying Information of Dynamical Biochemical Reaction Networks

**DOI:** 10.3390/e25060887

**Published:** 2023-06-01

**Authors:** Zhiyuan Jiang, You-Hui Su, Hongwei Yin

**Affiliations:** 1School of Science, Shenyang University of Technology, Shenyang 110870, China; 2School of Mathematics and Statistics, Xuzhou University of Technology, Xuzhou 221018, China; suyh02@163.com

**Keywords:** biochemical reaction networks, the length of information, information geometry

## Abstract

A large number of complex biochemical reaction networks are included in the gene expression, cell development, and cell differentiation of in vivo cells, among other processes. Biochemical reaction-underlying processes are the ones transmitting information from cellular internal or external signaling. However, how this information is measured remains an open question. In this paper, we apply the method of information length, based on the combination of Fisher information and information geometry, to study linear and nonlinear biochemical reaction chains, respectively. Through a lot of random simulations, we find that the amount of information does not always increase with the length of the linear reaction chain; instead, the amount of information varies significantly when this length is not very large. When the length of the linear reaction chain reaches a certain value, the amount of information hardly changes. For nonlinear reaction chains, the amount of information changes not only with the length of this chain, but also with reaction coefficients and rates, and this amount also increases with the length of the nonlinear reaction chain. Our results will help to understand the role of the biochemical reaction networks in cells.

## 1. Introduction

The basic unit of life is cells, and inside cells, a large number of biochemical reactions are constantly taking place. These reactions govern the differentiation of cells and the growth and development of individual organisms. They in general constitute a network of reactions that can regulate cellular behaviors. Recently, a large number of the biochemical reaction networks are identified for their functions. For instance, the signaling network of *Dpp* in *Drosophila* embryo [1,2,3,4] can regulate stem cell differentiation and control the rate of tissue growth; the signaling network of the P53 protein can protect and control cell cycle phase transitions [5,6,7,8]. In gene expression regulation networks, the transcription factors are recruited to the promoters, and biochemical reactions between them occur, resulting in activating the promoters [9,10,11,12]. The signaling network of JNK and P38 protein plays important roles in regulating cell inflammation and apoptosis, and can promote the apoptosis of cancer cells by activating these two information pathways [13,14,15,16,17]. The gene regulatory network of PTEN can regulate the synthesis of particular proteins and regulate the cell division cycle, inhibiting cancer cells [18,19,20]. Through phosphorylation, the signaling network of P13k-AKT forwardly or backwardly regulates substrate proteins to alter their intracellular localization or change their protein stability [21]. Aberrant activation of P13k-AKT causes uncontrolled cell proliferation and cell cycle dysregulation, inhibits apoptosis, and is associated with drug resistance in cancer therapy [22,23,24]. The regulatory network of CK2 can exert anti-apoptotic roles by protecting regulatory proteins from caspase-mediated degradation [25]. In the Xp10 phage regulatory network, P7 regulatory protein turns off host gene transcription or control the sequential expression of phage genes from early genes to late genes [26]. Therefore, studying the biochemical reaction networks is of great importance for understanding cell behaviors.

In fact, the biochemical reactions in cells are the processes of transmitting information between the upper and lower biochemical reactions [27,28], and the biochemical reaction networks in cells can transmit information from internal or external signals [29,30,31]. On the other hand, transmitting information through the biochemical reaction networks are also dynamical processes, possessing precise real-time information of controlling cell behaviors. For example, *Dpp* gradient always dynamically changes due to noise inside the embryos, but it is still precisely decoded by the downstream signaling pathway. A growing number of studies reveal that cells can encode and decode information by controlling temporal behavior of their signaling molecules [32]. This implies that the dynamical behavior of the signaling molecules can produce some information. How this information can be quantified is one of the hot topics in the field of system biology.

At present, information length, based on Fisher information and information geometry, is one method of measuring the information on dynamical behaviors. This method can explain the effect of different spatially periodic and deterministic forces [33]. Moreover, information length is linked with stochastic thermodynamics considering information geometry [34]. It can uncover the physical limitations in bacterial growth [35], and quantify the amount of information of morphogen gradient from its initial state to its steady state [36]. Here, we will apply this method to explore information of dynamical biochemical reaction networks, and measure the amount of information generated by the networks. This problem is, as far as we know, rarely studied.

In this paper, we mainly consider a simple biochemical reaction network by using the combination of information length and information geometry; we find that, in the linear biochemical reaction chain, with the increase of the reaction chain length, the amount of information does not always increase, and when the number of this length is not very large, the change of information transmission is larger. When this number reaches a critical value, the change of information transmission is less, and this phenomenon also confirms that some biochemical reaction networks are usually completed in 5–6 steps. For example, the degradation of mRNA and protein is only 2–3 steps, and the activation of the promoter is accomplished by regulating transcription factors in at most 8 steps [37,38]. In the nonlinear biochemical reaction chain, the change in the amount of information is not only related to the length of the reaction chain, but also to the reaction coefficients and the reaction rates. 

This paper is organized as follows. In Section 2, information geometry is introduced. In Section 3, we build respectively the linear and nonlinear biochemical reaction chains and explore the character of information transmitted in these two models. Finally, a brief discussion and conclusion are given.

## 2. Method of Information Geometry

In order to investigate the reaction network from the perspective of information geometry, we firstly introduce the information geometry on the discrete distribution. Due to the normalization of the probability distribution, a set of possible distributions can generate a Riemannian manifold. In this article, we only consider a set of discrete distribution group $p=\left(p_{0},p_{1},\cdots ,p_{n}\right),p_{i}\geq 0$ for any i, and $\sum_{i=0}^{n}p_{i}=1.$ We conventionally consider the Kullback–Leibler divergence between the two distributions p and $p^{\prime }$, defined as
(1)$$D_{kl}(p\|p^{\prime })=\sum_{i=0}^{n}p_{i}\ln\frac{p_{i}}{p_{i}^{\prime }}\cdot$$
In information geometry, the square of the line element ds is defined as the second-order Taylor series of the Kullback–Leibler divergence [39,40] between two probability distributions p and $p+dp=(p_{0}+dp_{0},p_{1}+dp_{1},\cdots {,p}_{n}+dp_{n})$,
(2)$$dl^{2}=\sum_{i=0}^{n}\frac{{\left(dp_{i}\right)}^{2}}{p_{i}},$$
where $dp=(dp_{0},dp_{1},\cdots ,dp_{n})$ is the infinitesimal displacement that satisfies $\sum_{i=0}^{n}dp_{i}=0$. The square of the line element is directly related to the Fisher information metric. Any probability distribution $p=\left(p_{0},p_{1},p_{2},\cdots ,p_{n}\right)$ corresponds with a point on the spherical surface by replacing the coordinate axis $p_{0},p_{1},p_{2},\cdots ,p_{n}$ with $2\sqrt{p_{0}},2\sqrt{p_{1}},2\sqrt{p_{2}},\cdots ,2\sqrt{p_{n}}$. Using this method, the simplex composed of the coordinate axis $p_{0},p_{1},p_{2},\cdots ,p_{n}$ turns into a manifold $S_{n}$ with an n-dimension spherical surface of radius 2 [17,33,34]. Next, we introduce the statistical length L [41,42,43],
(3)$$L=\int dl=\int \frac{dl}{dt}dt,$$
which can quantify the information change [40]. It is well known that the shortest of the curves connecting the two points on the spherical surface is the geodesic. The following relationship has been given out in [34,35].
(4)$$L\geq D\left(r_{ini}.r_{fin}\right)=2\arccos\left(r_{ini}.r_{fin}\right),$$
where “·” denotes the inner product; r is the unit vector defined as $r=\left(r_{0},r_{1},\cdots {,r}_{n}\right)=(\sqrt{p_{0}},\sqrt{p_{1}},\cdots ,\sqrt{p_{n}})$. $D\left(r_{ini},r_{fin}\right)$ stands for the geodesic (that is, the shortest arc) between $r_{ini}$ and $r_{fin}$ in the manifold $S_{n}$, and $arccos\left(r_{ini},r_{fin}\right)$ is the included angle between $r_{ini}$ and $r_{ini}$. Unlike the Kullback–Leibler divergence, the geodesic D is symmetric and satisfies the triangle inequality, so it can be treated as a true distance [41,44,45,46,47,48,49,50,51,52].

## 3. Model and Analysis

Here, we consider a general biochemical reaction chain(model 1),
(5)$$X_{1}\overset{k_{1}}{\underset{k_{-1}}{⇌}}n_{2}X_{2}\overset{k_{2}}{\underset{k_{-2}}{⇌}}n_{3}X_{3}\overset{k_{3}}{\underset{k_{-3}}{⇌}}n_{4}X_{4}\overset{k_{4}}{\underset{k_{-4}}{⇌}}\cdots \overset{k_{n-2}}{\underset{k_{-(n-2)}}{⇌}}n_{n-1}X_{n-1}\overset{k_{n-1}}{\underset{k_{-(n-1)}}{⇌}}n_{n}X_{n},$$
where $X_{1},X_{2},X_{3},X_{4},\cdots ,X_{n-1},X_{n}$ are reactants; $n_{2},n_{3},n_{4},\cdots ,n_{n-1},n_{n}$ are the reaction coefficient; $k_{1},k_{-1},k_{2},k_{-2},\cdots {,k}_{n-1},k_{-(n-1)}$ are the reaction rate constants. Here, we assume that the concentration of the first reactant is not zero and that the concentrations of the other reactants are zero, because only when the first reactant is involved in the reaction, the later reactants will occur.

For this kind of biochemical reaction network, in bacteria such as *E. coli* and *Salmonella*, and in eukaryotic cells such as Drosophila, their transcription processes are a cascade reaction as the above model, so it is interesting for us to study this reaction network.

### 3.1. Linear Biochemical Reaction Network

To uncover the underlying principle of information transformation via this network, we firstly consider a simple case of this network (model 2) by assuming the reaction coefficients to be 1, i.e.,
(6)$$X_{1}\overset{k_{1}}{\underset{k_{-1}}{⇌}}X_{2}\overset{k_{2}}{\underset{k_{-2}}{⇌}}X_{3}\overset{k_{3}}{\underset{k_{-3}}{⇌}}X_{4}\overset{k_{4}}{\underset{k_{-4}}{⇌}}\cdots \overset{k_{n-2}}{\underset{k_{-\left(n-2\right)}}{⇌}}X_{n-1}\overset{k_{n-1}}{\underset{k_{-\left(n-1\right)}}{⇌}}X_{n}.$$
Obviously, this model is a linear network of biochemical reactions, and is governed by the following master equation,
(7)$$\left\{\begin{array}{l} \frac{dp_{X_{1}\left(t\right)}}{dt}=k_{-1}p_{X_{2}\left(t\right)}-k_{1}p_{X_{1}\left(t\right)},\\ \frac{dp_{X_{2}\left(t\right)}}{dt}=k_{1}p_{X_{1}\left(t\right)}-k_{-1}p_{X_{2}\left(t\right)}+k_{-2}p_{X_{3}\left(t\right)}-k_{2}p_{X_{2}\left(t\right)},\\ \frac{dp_{X_{3}\left(t\right)}}{dt}=k_{2}p_{X_{2}\left(t\right)}-k_{-2}p_{X_{3}\left(t\right)}+k_{-3}p_{X_{4}\left(t\right)}-k_{3}p_{X_{3}\left(t\right)},\\ \;\;\;\vdots \\ \frac{dp_{X_{n}\left(t\right)}}{dt}=k_{n-1}p_{X_{n-1}\left(t\right)}-k_{-\left(n-1\right)}p_{X_{n}\left(t\right)}. \end{array}\right.$$
Because the large number of coupled equations in Equation (7) is larger, we hardly provide the analytic expression of the solution with respect to time t. Its corresponding steady state equation is
(8)$$\left\{\begin{array}{l} 0=k_{-1}p_{X_{2}\left(t\right)}-k_{1}p_{X_{1}\left(t\right)},\\ 0=k_{1}p_{X_{1}\left(t\right)}-k_{-1}p_{X_{2}\left(t\right)}+k_{-2}p_{X_{3}\left(t\right)}-k_{2}p_{X_{2}\left(t\right)},\\ 0=k_{2}p_{X_{2}\left(t\right)}-k_{-2}p_{X_{3}\left(t\right)}+k_{-3}p_{X_{4}\left(t\right)}-k_{3}p_{X_{3}\left(t\right)},\\ \;\;\;\vdots \\ 0=k_{n-1}p_{X_{n-1}\left(t\right)}-k_{-\left(n-1\right)}p_{X_{n}\left(t\right)}. \end{array}\right.$$
In help of the conservation rule of the probability, we can obtain its analytical solution,
(9)$$\left\{\begin{array}{l} p_{X_{1}}=\frac{k_{-1}k_{-2}k_{-3}k_{-4}\cdots k_{-\left(n-1\right)}}{k_{-1}k_{-2}k_{-3}k_{-4}\cdots k_{-\left(n-1\right)}+k_{1}k_{-2}k_{-3}k_{-4}\cdots k_{-\left(n-1\right)}+\cdots +k_{1}k_{2}k_{3}k_{4}\cdots k_{\left(n-1\right)}},\\ p_{X_{2}}=\frac{k_{1}}{k_{-1}}p_{X_{1}},\\ p_{X_{3}}=\frac{k_{1}k_{2}}{k_{-1}k_{-2}}p_{X_{1}},\\ \;\;\vdots \\ p_{X_{n}}=\frac{k_{1}k_{2}k_{3}k_{4}\cdots k_{\left(n-1\right)}}{k_{-1}k_{-2}k_{-3}k_{-4}\cdots k_{-\left(n-1\right)}}p_{X_{1}}. \end{array}\right.$$
This expression is a recursive formula.

### 3.2. Numerical Simulations and Results

In this section, we will numerically investigate information involved in the linear reaction Chain (See Equation (6)) from its initial state to its steady state. Since the reaction species in Model 2 (See Equation (6)) is usually a lot, numerical simulations are performed for Equation (7). In order to obtain general results, through this paper, the reaction rates in model 2(See Equation (6)) are randomly taken from the uniform distribution U (0.1,1), and then according to Equations (3) and (4), the geodesic curve and the information in Model 2 (See Equation (6)) from its initial state to its steady state can be, respectively, calculated numerically. This process is repeatedly performed for 2000 times. In Figure 1, we first show four geodesic curves of Model 2 (See Equation (6)), which are randomly chosen from the 2000-time random experiments. We can find that the length of the geodesic curves increases with the number of the reaction chain, and that the geodesics of the whole biochemical reaction will reach a stationary state when the number of the reactant species reaches a certain critical value. Of course, this critical value of the reaction chain is different for different reaction rates.

Next, the amount of information in Model 2 (See Equation (6)) from its initial state to its steady state will be examined. First, we choose 2000 groups of random numbers from the uniform distribution U (0.1,1). Each of these groups is settled as the reaction rates in Model 2 (See Equation (6)), and calculate the amount of information. We find that the amount of information in Model 2 (See Equation (6)) almost increases with the number of the reaction chain, see Figure 2. Although this amount may slightly drop once for the special reaction rates at a certain number of the linear reaction chain, the later amount of information will continuously increase, see Figure 3. To further uncover the dropping range of the information amount, we count the statistical frequency of the drop in the information amount, see Figure 4. This figure shows the frequencies for less dropping ranges, and that the frequency of the dropping range of 0.5 or more counts for 18.3%, implying the amount of the information may always increase if calculation errors are ignored. Of course, through 2000 random simulations, we find that the critical values, at which the amount of information in the reaction chain reaches stationary states, are different when the forward and backward reaction rates are different.

### 3.3. Nonlinear Biochemical Reaction Network

Next, we obtain the nonlinear network of biochemical reactions, while reaction coefficients are not all one (model 3).
(10)$$X_{1}\overset{k_{1}}{\underset{k_{-1}}{⇌}}n_{2}X_{2}\overset{k_{2}}{\underset{k_{-2}}{⇌}}n_{3}X_{3}\overset{k_{3}}{\underset{k_{-3}}{⇌}}n_{4}X_{4}\overset{k_{4}}{\underset{k_{-4}}{⇌}}\cdots \overset{k_{n-2}}{\underset{k_{-\left(n-2\right)}}{⇌}}n_{n-1}X_{n-1}\overset{k_{n-1}}{\underset{k_{-\left(n-1\right)}}{⇌}}n_{n}X_{n}.$$

Its master equation is
(11)$$\left\{\begin{array}{l} \frac{dp_{X_{1}\left(t\right)}}{dt}=k_{-1}p_{X_{2}\left(t\right)}^{n_{2}}-k_{1}p_{X_{1}\left(t\right)},\\ \frac{dp_{X_{2}\left(t\right)}}{dt}=k_{1}p_{X_{1}\left(t\right)}-k_{-1}p_{X_{2}\left(t\right)}^{n_{2}}+k_{-2}p_{X_{3}\left(t\right)}^{n_{3}}-k_{2}p_{X_{2}\left(t\right)}^{n_{2}},\\ \frac{dp_{X_{3}\left(t\right)}}{dt}=k_{2}p_{X_{2}\left(t\right)}^{n_{2}}-k_{-2}p_{X_{3}\left(t\right)}^{n_{3}}+k_{-3}p_{X_{4}\left(t\right)}^{n_{4}}-k_{3}p_{X_{3}\left(t\right)}^{n_{3}},\\ \;\;\;\vdots \\ \frac{dp_{X_{n}\left(t\right)}}{dt}=k_{n-1}p_{X_{n-1}\left(t\right)}^{n_{n-1}}-k_{-\left(n-1\right)}p_{X_{n}\left(t\right)}^{n_{n}}. \end{array}\right.$$

Since, the equation is a nonlinear coupled master equation, we cannot obtain its analytical solution; we can only perform some numerical results via random simulation and calculate the probability of each reactant. Thus, we can obtain the amount of information through the method of information geometry. In Figure 5, we consider a nonlinear biochemical reaction network with the chain length 3 (model 4), as follows:(12)$$X_{1}\overset{k_{1}}{\underset{k_{-1}}{⇌}}mX_{2}\overset{k_{2}}{\underset{k_{-2}}{⇌}}nX_{3},$$
and plot the heat map of the information for Model 4 (See Equation (12)), which finds that it increases with the number of reactants involved in the reaction.

In Figure 5, we find that the chain reaction is most informative when m=10 and n=10. In the following, we will increase the length of the reactants and fix m=10,n=10, change the size of $m_{1},n_{1}$ behind, and observe when the combination of reactants in the chain reaction is the maximum amount of information. Thus, we will consider a nonlinear biochemical reaction network with a chain length of 5 (model 5)as follows:(13)$$X_{1}\overset{k_{1}}{\underset{k_{-1}}{⇌}}mX_{2}\overset{k_{2}}{\underset{k_{-2}}{⇌}}nX_{3}\overset{k_{2}}{\underset{k_{-2}}{⇌}}{m_{1}X}_{4}\overset{k_{2}}{\underset{k_{-2}}{⇌}}{n_{1}X}_{5},$$
and plot the heat map of the information for Model 5 (See Equation (13))

In Figure 6, we find that the reaction chain has the most information when $m_{1}=4$ and $n_{2}=1.$ We perform another 2000 random simulation experiments on the nonlinear network Model 3 (See Equation (10)), where the reaction coefficients are also randomly chosen from the uniform integer distribution in the range [1, 10], and we randomly select four random simulations to show the results in Figure 7. In Figure 7, we can find that in Model 3 (See Equation (10)), the amount of information increases with the length of the nonlinear reaction chain, and hardly changes after a certain critical length of the chain. Such a phenomenon is presented in all these 2000 random simulation experiments.

Metabolic networks need to contain multiple cascades of enzyme-catalyzed reactions, and the substrate and product of each reaction are a substrate or product of another reaction. In this case, the interdependence between these catalytic reactions will lead to a nonlinear kinetic behavior of the metabolic network.

A good example is the enzyme-catalyzed reactions in the glycogen metabolic network, where multiple cascades of enzyme-catalyzed reactions lead to the synthesis and degradation of glycogen. In addition, many other metabolic networks, such as the glucose metabolic network, the fatty acid metabolic network, and the amino acid metabolic network, also fit this condition. The enzyme-catalyzed reactions in each of these metabolic networks can be described using the above results to better understand their nonlinear kinetic behavior.

## 4. Conclusions

In conclusion, studying biochemical reaction networks is of great importance for understanding cellular behaviors, and the networks regulating cellular differentiation, growth, and development. Biochemical reaction networks are signaling pathways that transmit information, and the dynamic behavior of signaling molecules can generate information that can be quantified by using information length based on Fisher information. In this paper, we explore the amount of information generated via linear and nonlinear biochemical reaction networks. We find that in linear networks, the amount of information does not increase always with the length of the reaction chain; instead, the amount of information varies significantly when the number the reactant participation is not very large. When the number of reactants involved reaches a critical value, the amount of information rarely changes. This phenomenon confirms that some biochemical reaction networks are usually completed in 5–6 steps. In nonlinear networks, the change in the amount of information is not only related to the participation of reactants, but also to the combination of reaction coefficients and the change in the reaction rate. 

It is worth noting that our results are novel and different from previous results, which showed that the reaction cascade can strengthen noise [53]. On the other hand, our model is simple and does not consider complex cases, such as the feedback in our model and polymerization reactions and so on. These problems will be investigated in our future works. 

## Figures and Tables

**Figure 1 entropy-25-00887-f001:**
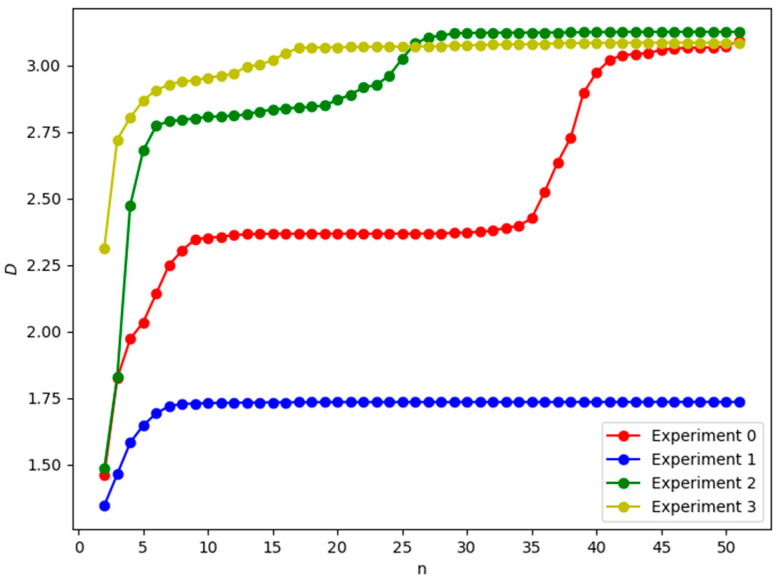
The geodesic of Model 2 (See Equation (6)). *n* indicates the number of the reaction chain in the linear network, and *D* indicates the geodesic curve, where the rates are shown in Table A1 in Appendix A.

**Figure 2 entropy-25-00887-f002:**
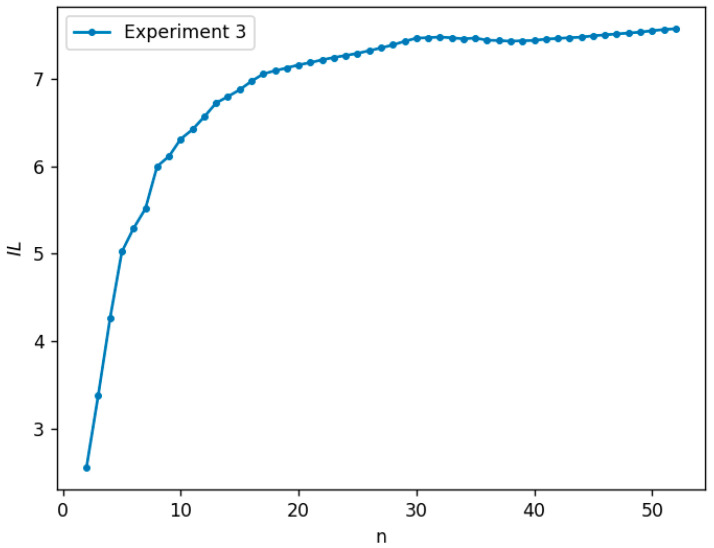
The amount of information of Model 2 (See Equation (6)). *n* represents the length of the chains in the linear network, and *IL* represents the amount of information.

**Figure 3 entropy-25-00887-f003:**
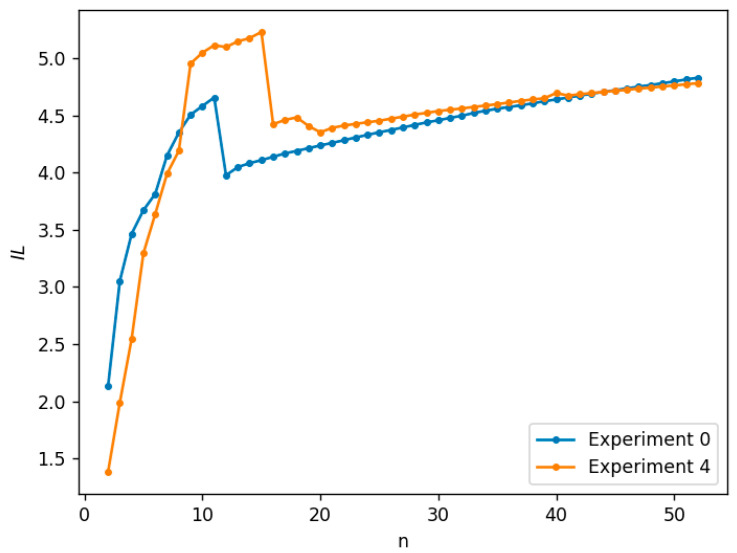
Information scatter plot of the linear biochemical Reaction Network Equation (6). *n* represents the length of the chains in the linear network, and *IL* represents the amount of information. The reactant coefficients and reaction rates for each randomized simulation experiment are shown in Table A1, in Appendix A.

**Figure 4 entropy-25-00887-f004:**
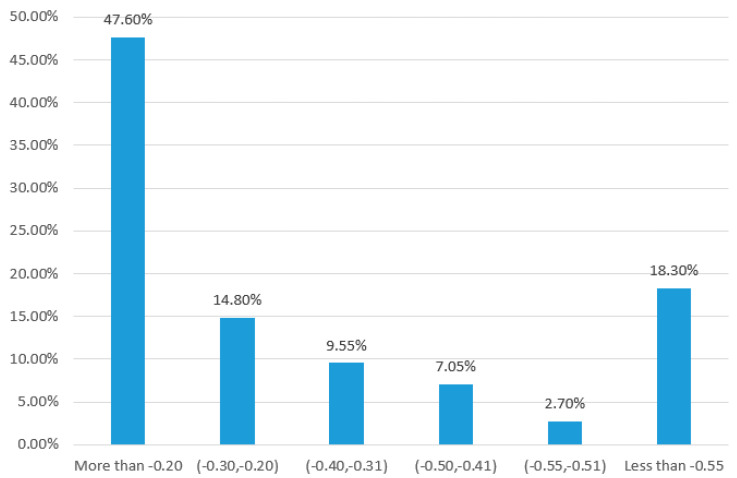
Statistical chart of dropping in the information in different ranges. Calculating the information amount in Model 2 (See Equation (6)) is performed for 2000 times by choosing random values from U (0.1, 1) as the reaction rates.

**Figure 5 entropy-25-00887-f005:**
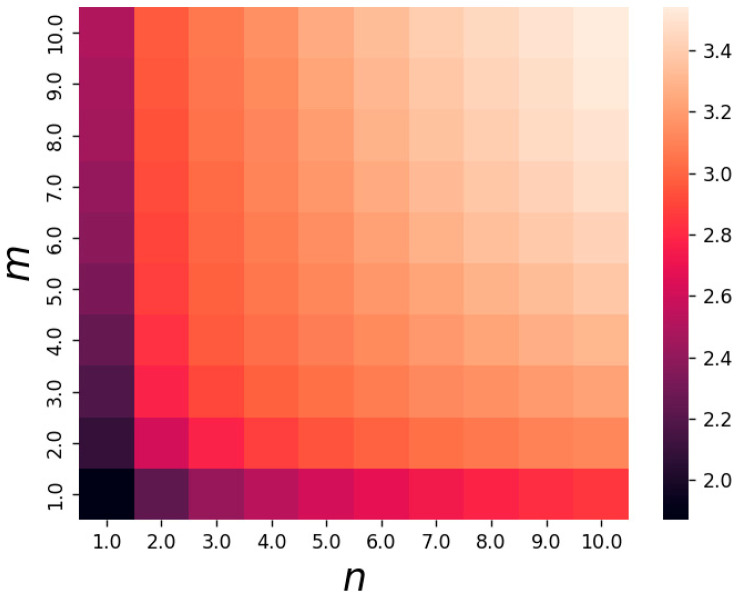
The heat map of information. m,n are the coefficients of $X_{2},X_{3}$ reactants, and $k_{1}=0.21,k_{2}=0.32,k_{-1}=0.56,k_{-2}=0.67$. Brighter color in the diagram implies more information.

**Figure 6 entropy-25-00887-f006:**
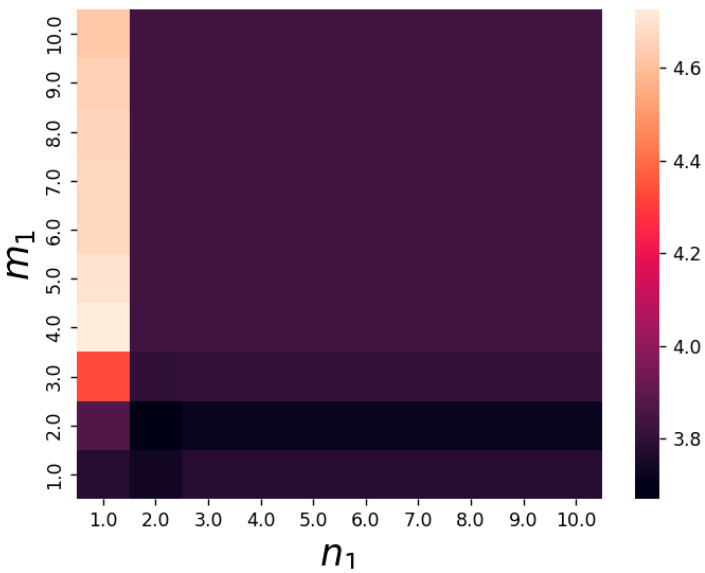
The heat map of information. $m_{1},n_{1}$ are the coefficients of $X_{4},X_{5}$ reactants, and $k_{1}=0.21,k_{2}=0.32,{k_{3}=0.1{,k}_{4}=0.2,k}_{-1}=0.56,k_{-2}=0.67{,k}_{-3}=0.92{,k}_{-4}=0.8$. The brighter color in the diagram implies more information.

**Figure 7 entropy-25-00887-f007:**
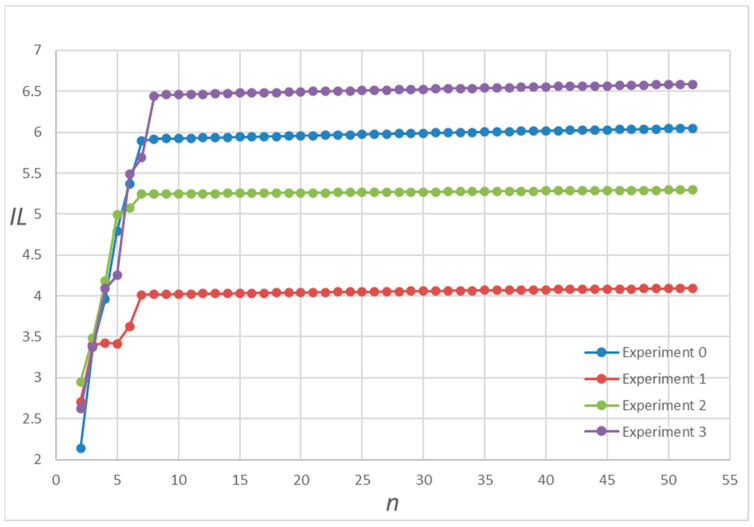
Information scatter plot of the nonlinear biochemical Reaction Network (Equation (10)). *n* represents the length of the chains in the linear network, and *IL* represents the amount of information. The reactant coefficients and reaction rates for each randomized simulation experiment are shown in Table A1 and Table A2 in Appendix A.

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
