# Peer review of "Quantifying Information of Dynamical Biochemical Reaction Networks"

_entropy, 2023, doi:10.3390/e25060887_

Round 1

Reviewer 1 Report

In this manuscript, the authors calculated the information length and the geodesic of two reaction network models. One model is linear, in which the reaction coefficient is 1. The other model is a two-step reaction. While the models are so simplified and calculation work is not extensive, exploring information of dynamical biochemical reaction networks is worth encouraging. I recommend its publication.

The following are some questions and comments:

1.       The two models are too ideal. In the first model, the forward reaction rates are all the same for up to 20 reactions in the network, and so are all the backward reaction rates. Too simplified models may provide little insight into real-world problems.

2.       In the introduction, the authors listed a large number of biochemical reactions that play important roles in life. But there is no detailed discussion of what their calculation result means in cells. Maybe the authors can choose one real biochemical reaction network, such as P53 signaling, and use real reaction coefficient and reaction rate to perform calculations, and then discuss what new things they learn from the calculation results.

3.       Calculation details are missing. Basically, the authors solved ordinary differential equations. What algorithm was used? Runge-Kutta? Which order? What are the initial values? What software was used? There should be a method section talking about these details.

4.       In Eq 2.4, there should be a dot instead of a comma.

Reviewer 2 Report

The authors define a geodesic measure, D, for the amount of information, IL, in a probability distribution. They apply these concepts to a set of linear and nonlinear sequential reactions. The main take-away of the work is that the geodesic and amount of information seem to be “saturated” after a certain number of reactions in the sequence and lengthening the sequence after that does not change the amount of information. This can be important when activating promotors by regulating transcription factors in cells.  

The introduction of the manuscript is well written and the principles behind the idea that the authors have is interesting and logically presented. The part that needs more work are the results, which even within the limited scope of the examples given here are rather sparse. The authors consider only one set of reaction rates for the linear reaction system. Can the authors prepare more combinations of reaction rates possible and show they show the same saturation behaviour? The authors affectively use different values of k1 in Figure 3, but still use the value of 0.2 for all the other reaction rates. If the reaction rates are not all equal, the saturation effect may not arise. Working out more details on smaller series of reactions (e.g., n = 2 or 3) with different combinations of ki and then generalizing may be one way to proceed.

Furthermore, can the authors add more chemical / biological content to the results of the non-linear sequence they discuss in Figure 4.

There are cases of sequential chemical reactions in non-biological systems. Can the authors link their results to studies of these systems as well?

In page 3, line 107, the authors define the “.” for the inner product, but this does not seem to be used in Eq. (2.4).

Reviewer 3 Report

It is an application of a mathematical method for calculation (numerical simulation) the information entropy for a simple decomposition reaction. The conclusion is clear, but It is not clear to me what can we learn from it. It is necessary to a more clear explanation for that which type of biological network can be described with this equatuion. Additionally I suggest to put some sentence about the results for nonlinear equation.

Author Response

请参阅附件。

Round 2

Reviewer 2 Report

The authors have addressed my comments and the additional material improves the content of the manuscript. I can now recommend publication of this work.

Reviewer 3 Report

In the present form the paper is acceptable. I can accept all of the answers.